



# AIEADA 1.0: Efficient high-dimensional variational data assimilation with machine-learned reduced-order models

Romit Maulik[1], Vishwas Rao[1], Jiali Wang[1], Gianmarco Mengaldo[2], Emil Constantinescu[1], Bethany Lusch[1], Prasanna Balaprakash[1], Ian Foster[1], and Rao Kotamarthi[1]

[1]Argonne National Laboratory
[2]National University of Singapore

**Correspondence:** Romit Maulik (rmaulik@anl.gov)

**Abstract.** Data assimilation (DA) in the geophysical sciences remains the cornerstone of robust forecasts from numerical models. Indeed, DA plays a crucial role in the quality of numerical weather prediction, and is a crucial building block that has allowed dramatic improvements in weather forecasting over the past few decades. DA is commonly framed in a variational setting, where one solves an optimization problem within a Bayesian formulation using raw model forecasts as a prior, and observations as likelihood. This leads to a DA objective function that needs to be minimized, where the decision variables are the initial conditions specified to the model. In traditional DA, the forward model is numerically and computationally expensive. Here we replace the forward model with a low-dimensional, data-driven, and differentiable emulator. Consequently, gradients of our DA objective function with respect to the decision variables are obtained rapidly via automatic differentiation. We demonstrate our approach by performing an emulator-assisted DA forecast of geopotential height. Our results indicate that emulator-assisted DA is faster than traditional equation-based DA forecasts by four orders of magnitude, allowing computations to be performed on a workstation rather than a dedicated high-performance computer. In addition, we describe accuracy benefits of emulator-assisted DA when compared to simply using the emulator for forecasting (i.e., without DA). Our overall formulation is denoted AIAEDA (Artificial Intelligence Emulator Assisted Data Assimilation)

## 1 Introduction

A physical system can be characterized by our existing knowledge of the system plus a set of observations. Existing knowledge of the system is typically formulated in terms of a mathematical model (hereafter, also referred to as *model* or *computational model*) that usually consists of differential equations. Observations can arise from various sensors, both remote (e.g., satellites) and in-place (e.g., weather stations and radiosondes).

Data assimilation (DA) combines existing knowledge of a system, usually in the form of a model, with observations to infer the best estimate of the system state at a given time. Both existing knowledge and observations come with errors that lead to uncertainties about the "true" state of the system being investigated. Hence, when combining *model* results characterizing our knowledge of the system with observations, it is essential to account for these errors and give an appropriate weight to each source of information available. This leads to statistical approaches, which are the basis for state-of-the-art DA methods.





Model results and observations, along with their uncertanties, are encapsulated within a Bayesian framework to provide the
best estimate of the state of the system conditioned on the model and observation uncertainties (or error statistics) Daley (1993);
Kalnay (2003); Le Dimet and Talagrand (1986). More specifically, DA can be formulated as a class of inverse problems that
combines information from: 1) an *uncertain prior* (also referred to as *background*) that encapsulates our best estimate of the
system at a given time; 2) an *imperfect computational model* describing the system, and 3) *noisy observations*. These three
sources of information are combined together to construct a posterior probability distribution that is regarded as the best
estimate of the system state at the given time(s) of interest and is referred to as *analysis*. The analysis can be used for various
tasks, including optimal state identification, and selection of appropriate initial conditions for computational models.

Two approaches to DA have gained widespread popularity: variational and ensemble-based estimation methods Kalnay
(2003). The former are derived from optimal control theory, while the latter are rooted in statistics. Variational methods formu-
late DA as a constrained nonlinear optimization problem. Here, the state of the system is adjusted to minimize the discrepacy
between the prior or exisiting knowledge (e.g., in the form of a computational model) and observations, where their associ-
ated error statistics are commonly prescribed. Ensemble-based methods use optimal statistical interpolation approaches, and
error statistics for the prior and observations are obtained from an ensemble. Regardless of the approach adopted, DA can
be performed in two ways: sequential and continuous. In the sequential way, observations are assimilated in batches as they
become available. In the continuous way, one defines a prescribed time window, called *assimilation window*, and uses all the
observations available within the window to obtain the analysis.

DA, originally developed for numerical weather prediction (NWP), can be traced back to the original work of Richard-
son Lynch (2008). Today, DA is used extensively in NWP to compute accurate states of the atmosphere that, in turn, are used
to estimate appropriate initial conditions for NWP models, to compute reanalysis, and to help better understand properties
of the atmosphere. We focus here on NWP applications, although the novel methodology proposed can be applied in other
contexts.

In NWP, variational approaches are the workhorse, and have been employed for the past several years by leading operational
weather centers, including the European Centre for Medium-Range Weather Forecasts (ECWMF) and National Centers for
Environmental Prediction (NCEP). The two main methods adopted are three-dimensional variational DA (or 3D-Var), and
four-dimensional variational DA (or 4D-Var). In both cases, one defines an assimilation window and seeks the system state
that best fits the data available, comprising the prior and observations. In 3D-Var, all observations are regarded as if they were
obtained at the same time snapshot (i.e., there is no use of the time dimension of the assimilation window). In 4D-Var, the
observations retain their time information, and one seeks to identify the state evolution (also referred to as the trajectory) that
best fits them within the assimilation window. The state evolution is commonly obtained via a computational model, consisting
of a set of high-dimensional partial differential equations (PDEs), that is often considered perfect (without errors) within the
assimilation window.

The current state-of-the-art is 4D-Var. This is often implemented as a strong constraint algorithm (SC4D-Var) where one
assumes that the observations over the time window are consistent (within a margin of observation errors) with the model
if initialized by the true state, where the model is considered to be perfect. In reality, the error in the model is often non-




negligible, in which case the SC4D-Var scheme produces an analysis that is inconsistent with the observations. The effect of
model error is even more pronounced when the assimilation window is large. Weak constraint 4D-Var (WC4D-Var) Trémolet
(2006, 2007) relaxes the 'perfect model' assumption and assumes that model error is present at each time snapshot of the
assimilation window. This requires estimates of model-error statistics that are commonly simplistic Brajard et al. (2020). Yet,
recent efforts have focused on improving estimates for model-error statistics and on better understanding their impact on the
analysis accuracy in both variational and statistical approaches Akella and Navon (2009); Cardinali et al. (2014); Hansen
(2002); Rao and Sandu (2015); Trémolet (2006, 2007); Zupanski and Zupanski (2006). Indeed, there has been a push to
aggregating computational NWP model uncertainties, such as those due to incomplete knowledge of the physics associated
with sub-grid modeling, errors in boundary conditions, accumulation of numerical errors, and inaccurate parametrization of
key physical processes, into a component called *model error* Glimm et al. (2004); Orrell et al. (2001); Palmer et al. (2005).

It has been shown that 4D-Var systematically outperforms 3D-Var Lorenc and Rawlins (2005), and for this reason is today
the state-of-the-art DA for NWP applications. However, the better accuracy of 4D-Var comes with the price of higher com-
putational costs. Indeed, to exploit the time dimension of the assimilation window it is necessary to repeatedly solve both the
computational model forward in time and the tangent linear and adjoint problems Errico (1997); Errico et al. (1993); Errico
and Raeder (1999). These two additional steps are particularly expensive and lead to a significantly larger computational cost
for the 4D-Var algorithm compared to its 3D-Var counterpart. Hence, a significant fraction of the computational cost in NWP
is due to DA. This cost can be equivalent to the cost of 30–100 model forecasts, which corresponds to the number of iterations
in the optimization procedure. This high cost restricts the amount of data that can be assimilated, and thus only a small fraction
of the available observations are typically employed for operational forecasts Bauer et al. (2015); Gustafsson et al. (2018).

Our goal in this work is to alleviate the computational burden associated with the 4D-Var approach by replacing the expen-
sive computational model and its adjoint with a data-driven emulator. (We use the terms emulator and surrogate interchangeably
in this document.) Because our emulator is easily differentiable, we can use automatic differentiation (AD) to avoid solving
the expensive adjoint problem. AD, in contrast to numerical differentiation, does not introduce any discretization errors such
as those encountered in finite differences. The lack of discretization-based gradient computation also leads to accurate compu-
tation of higher-order derivatives where such errors are more pronounced. Morever, when gradients are needed with respect to
many inputs (such as for partial differentiation), AD is more computationally efficient. Unlike numerical and symbolic differ-
entiation, AD relies on the chain rule to decompose differentials and compute them efficiently. Therefore, they are instrumental
in computing gradients with respect to inputs or parameters in neural network applications where the chain rule may be imple-
mented trivially. Therefore, much of the computational burden associated with the 4D-Var method is alleviated by replacing
the expensive computational models consisting of high-dimensional PDEs with machine learning (ML)-based surrogates.

A few efforts toward the integration of DA and ML have been recently undertaken. In Brajard et al. (2021), rather than
replacing an entire physics model with an ML emulator, DA is applied to an imperfect physics model, and ML is used to predict
the model error. In Frerix et al. (2021), ML is used to improve DA by learning a mapping from observational data to physical
states. Guen and Thome (2020) proposes an ML approach to video prediction that includes a novel neural network cell inspired
by DA with component called a "correction Kalman gain". In Hatfield et al. (2021), the original physical parametrization



scheme is not replaced by an emulator, but neural networks are used to derive tangent-linear and adjoint models to be used during 4D-Var. Mack et al. (2020) presents a new formulation for 3D-Var DA that uses convolutional autoencoders to create a reduced space in which to perform DA. In contrast to our work, they do not create an emulator to step forward in time, as they are performing 3D-Var DA, not 4D-Var. In Brajard et al. (2020), the authors use a method that iterates between training an ML surrogate model for a Lorenz system and applying DA. The output analysis then becomes the new training data to further improve the surrogate. In Penny et al. (2021), they train ML emulators for Lorenz models using a form of RNNs based on reservoir computing. Then DA is applied (4D-Var and the ensemble transform Kalman filter), estimating the forecast error covariance matrix using their RNN and deriving the corresponding tangent linear model and its adjoint as linear operators. In Casas et al. (2020), a recurrent neural network is trained to predict the difference between model outputs and a-priori performed DA computations during forecasting in a low-dimensional subspace spanned by truncated principal components. Similar methods may also be found in Pawar et al. (2020); Pawar and San (2021); Popov and Sandu (2021), where ML surrogates are used in lieu of the expensive forward model in ensemble techniques.

Our study is unique in several manners. First, we demonstrate the use of ML emulators to improve variational DA through enabling an acceleration of the outer-loop optimization problem. By using a differentiable ML surrogate instead of an expensive numerical solver, rapid computation of gradients via automatic differentiation allows for a speed-up of several orders of magnitude. Morever, in contrast with Penny et al. (2021) where a reservoir computer was used as the surrogate, our ML emulator is given by deep recurrent neural network (i.e., a long short-term memory neural network with several stacked cells) for which automatic differentiation is imperative. Our formulation also employs a model-order reduction methodology to forecast dynamics on a reduced space, thereby leading to significant computational gains even for forecasting very high dimensional systems. In contrast with with the formulation in Casas et al. (2020), we perform DA 'on-the-fly' during forecasting instead of learning the mismatch between model predictions and DA-corrected values. This makes our forecasts more generalizable during testing conditions. The specific contributions of this study are summarized as:

– We propose a differentiable reduced-order surrogate model using dimensionality reduction coupled with a data-driven time-series forecasting technique.

– We construct a reduced-order variational DA optimization problem that updates the initial condition of any forecast, given observations from random sensors.

– We accelerate this DA by several orders of magnitude by using gradients of the differentiable low-order surrogate model. Our a-posteriori assessments show that the reduced-order DA significantly improves the accuracy of the forecast using the updated initial conditions.

We highlight that in contrast to a vast majority of previous DA and ML studies, the proposed technique is demonstrated for a forecasting problem that is of real-world importance (geopotential height). Our overall formulation, AIEADA (Artificial Intelligence Emulator Assisted Data Assimilation) will comprise a code base that will ultimately develop emulators and data assimilation for climate/weather models.





## 2   Surrogate modeling

In this section, we will first introduce our surrogate model strategy, which may be used for direct forecasting of a geophysical quantity from data. Subsequently, we introduce the DA procedure for real-time forecasting correction using this surrogate
model. The surrogate model relies on a dimensionality reduction given by proper orthogonal decomposition and neural network time-series forecasting of the reduced representation. We review dimensionality reduction, time-series forecasting, and surrogate-based DA in the following.

### 2.1   Proper orthogonal decomposition

The first step in the surrogate construction is to reduce the degrees of freedom of the original data set to enable rapid forecasts.
Projection-based reduced order models (ROMs) are effective at compressing high-dimensional dynamical systems without loss of spatio-temporal structure. The compression is performed by projecting the high-dimensional model onto a set of optimally chosen basis functions with respect to the $L_2$ norm Berkooz et al. (1993). This process can be illustrated for a state variable $\mathbf{x} \in \mathbb{R}^N$, where $N$ represents the size of the computational grid. Here, we use a project to approximate $\mathbf{x}$ as the linear expansion on a finite number of $k$ orthonormal basis vectors $\boldsymbol{\phi}_i \in \mathbb{R}^N$, a subset of the *POD basis*:

$$140 \quad \mathbf{x} \approx \sum_{i=1}^{k} r_i \boldsymbol{\phi}_i, \tag{1}$$

where $r_i \in \mathbb{R}$ is the $i$th component of $\mathbf{r} \in \mathbb{R}^k$, which are the coefficients of the basis expansion. The $\{\boldsymbol{\phi}_i\}$, $i = 1, \ldots, k$, $\boldsymbol{\phi}_i \in \mathbb{R}^N$ are the POD *modes*. POD modes in Equation 1 can be shown to be the left singular vectors of the snapshot matrix (obtained by stacking $M$ snapshots of $\mathbf{x}$), $\mathbf{X} = [\mathbf{x}_1, \ldots, \mathbf{x}_M]$, extracted by performing a singular value decomposition (SVD) on $\mathbf{X}$ Holmes et al. (2012); Chatterjee (2000). That is,

$$145 \quad \mathbf{X} \underset{\text{svd}}{=} \mathbf{U} \boldsymbol{\Sigma} \mathbf{V}^\top, \tag{2}$$

where $\mathbf{U} \in \mathbb{R}^{N \times M}$ and $\boldsymbol{\Phi}_k$ are the first $k$ columns of $\mathbf{V}$ after truncating the last $M - k$ columns based on the relative magnitudes of the cumulative sum of their singular values (see e.g., Maulik and Mengaldo (2021)). The total $L_2$ error in approximating the snapshots via the truncated POD basis is then

$$\sum_{j=1}^{M} \left\| \mathbf{x}_j - (\boldsymbol{\Phi}_k \boldsymbol{\Phi}_k^\top) \mathbf{x}_j \right\|_2^2 = \sum_{i=k+1}^{M} \sigma_i^2, \tag{3}$$

where $\sigma_i$ is the singular value corresponding to the $i$th column of $\mathbf{V}$ and is also the $i$th diagonal element of $\boldsymbol{\Sigma}$. It is well known that POD bases are $L_2$-optimal and present a good choice for efficient compression of high-dimensional data Berkooz et al. (1993). A recent alternative for compressing the data consists of spectral POD Schmidt et al. (2019); Mengaldo and Maulik (2021); Lario et al. (2021).





## 2.2 Time-series forecasting

For forecasting a dynamical system from examples of time-series data, we utilize an encoder-decoder framework coupled with long short-term memory neural networks (LSTMs) Hochreiter and Schmidhuber (1997). The encoder-decoder formulation is given by a first step where a latent representation is derived through information from historical observations (in our case, the compressed representation of the full state), i.e.,

$$\mathbf{h} = h(\mathbf{r}_{t-T}, \mathbf{r}_{t-T+1}, \ldots, \mathbf{r}_t) \tag{4}$$

where $h$ is the output of a function approximated by an LSTM. The LSTM neural architecture is devised to account for long and short-term correlations in time-series data through the specification of a hidden state that evolves over time and is affected by each observation of the dynamical system. The value of the hidden state at the end of the input sequence of length $T$ may then be employed as an encoded representation $\mathbf{h}$ of the input window. After $\mathbf{h}$ is obtained, forecasts at different distances in the future may be obtained by functional predictions. In this study, the 'decoding' component of the architecture is also given by an LSTM cell that is provided the encoded state information for each timestep of the output, i.e.,

$$[\mathbf{r}_{t+1}, \ldots, \mathbf{r}_{t+T_o}] = \tilde{h}(\boldsymbol{h}_{t+1}, \ldots, \boldsymbol{h}_{t+T_o}), \tag{5}$$
$$\boldsymbol{h}_{t+1} = \boldsymbol{h}_{t+2} = .. = \boldsymbol{h}_{t+T_o} = \boldsymbol{h}$$

where $\tilde{h}$ is an LSTM cell as well and $T_o$ is the length of the output window. Once forecasts in the POD-compressed space are obtained using the encoded-decoder LSTM network, the full-order state variable can be reconstructed by using the precomputed basis functions. A schematic of the architecture is shown in Figure 1. In the past several dynamical systems forecasts have been performed solely with the use of POD-LSTM type learning Pawar et al. (2019); Mohan and Gaitonde (2018); Maulik et al. (2021, 2020). However, as will be demonstrated shortly, this approach can be enhanced significantly with the use of real-time DA from sparse observations.

## 3 Data Assimilation

Data assimilation (DA) is the process of combining information from prior data, imperfect model predictions, and noisy observations to obtain an improved description of the true state $\mathbf{x}^{\text{true}}$ of a physical system. The resulting estimate represents the maximum a posteriori estimate, within a Bayesian setting, and is referred to as the analysis $\mathbf{x}^{\text{a}}$.

The *prior* represents the current knowledge of the system and is frequently referrred to as background in the DA community Kalnay (2003). The background is usually an estimate of the state $\mathbf{x}^{\text{b}}$, along with the corresponding error covariance matrix $\mathbf{B}$.

The *imperfect predictions* are generated by a model that approximates the physical laws that govern system evolution. The model evolution uses an initial state $\mathbf{x}_0 \in \mathbb{R}^n$ at initial time $t_0$ to obtain states $\mathbf{x}_i \in \mathbb{R}^n$ at future times $t_i$, i.e.,

$$\mathbf{x}_i = \mathcal{M}_{t_0 \to t_i}(\mathbf{x}_0), \quad i = 1, \cdots, N. \tag{6}$$





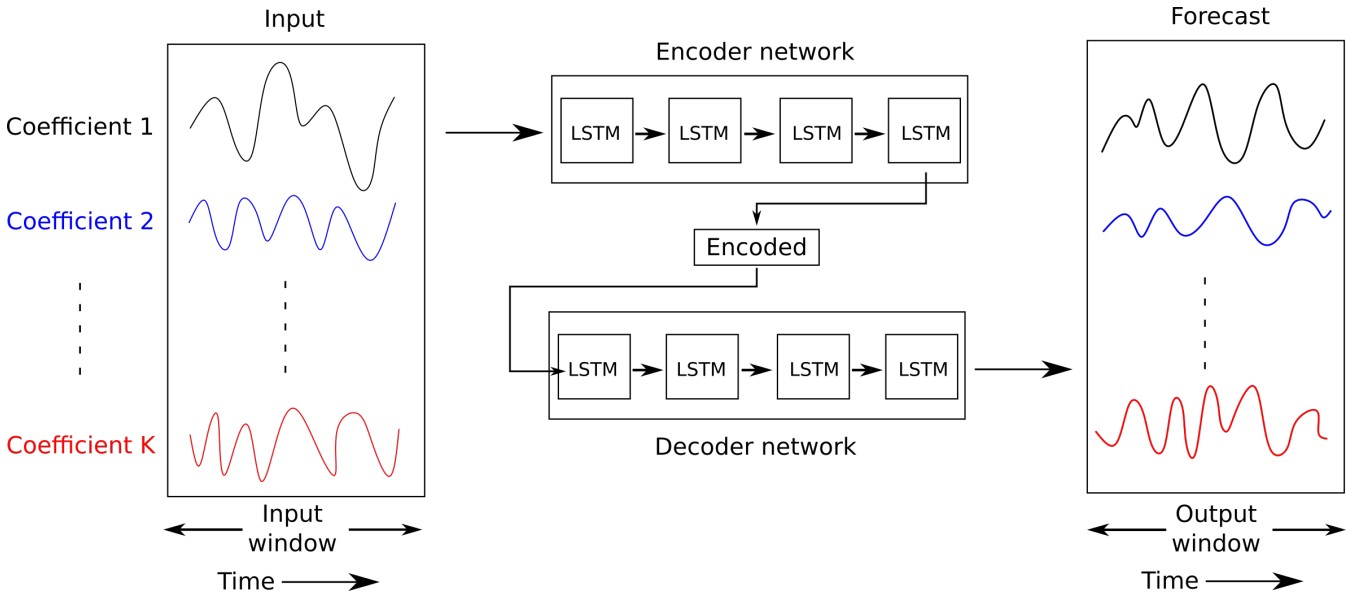

**Figure 1.** A schematic for forecasting POD coefficients using an encoder-decoder LSTM neural network. The various POD coefficients of an input window are encoded to a latent state on which the forecast window is conditioned.

The *noisy data* are partial observations of the true state available at discrete time instances. Specifically, measurements $\mathbf{y}_i \in \mathbb{R}^m$ of the true physical state $\mathbf{x}^{\text{true}}(t_i)$ are taken at discrete times $t_i$,

$$\mathbf{y}_i = \mathcal{H}(\mathbf{x}_i) + \varepsilon_i, \quad \varepsilon_i \sim \mathcal{N}(\mathbf{0}, \mathbf{R}_i), \quad i = 1, \cdots, N, \tag{7}$$

where the observation operator $\mathcal{H} : \mathbb{R}^n \rightarrow \mathbb{R}^m$ maps the high-dimensional model state space onto the partial (and potentially sparse) observation space. The random observation errors $\varepsilon_i$ are assumed to be normally distributed. In general, both the model and the observation operator are nonlinear. These concepts are detailed in the following referencs Daley (1993); Kalnay (2003); Sandu and Chai (2011); Sandu et al. (2005); Carmichael et al. (2008).

Through variational methods, one may solve the DA problem by adjusting a control variable (e.g., model parameters or initial conditions) in order to minimize the discrepancy between model forecasts and observations, in a manner similar to an optimal control framework. We next review the 4D-Var formulation.

### 3.1 Four-dimensional variational data assimilation

Strong-constraint four-dimensional variational (4D-Var) DA processes simultaneously all observations at all times $t_1, t_2, \ldots, t_T$ within the assimilation window (see 2 for a schematic representation). The control parameters are typically given by the initial conditions $\mathbf{x}_0$, which uniquely determine the state of the system at discrete instances in the future under the assumption that the model (6) perfectly represents reality. The background state (i.e., the prior) is the best estimate of the initial conditions $\mathbf{x}_0^{\text{b}}$, and





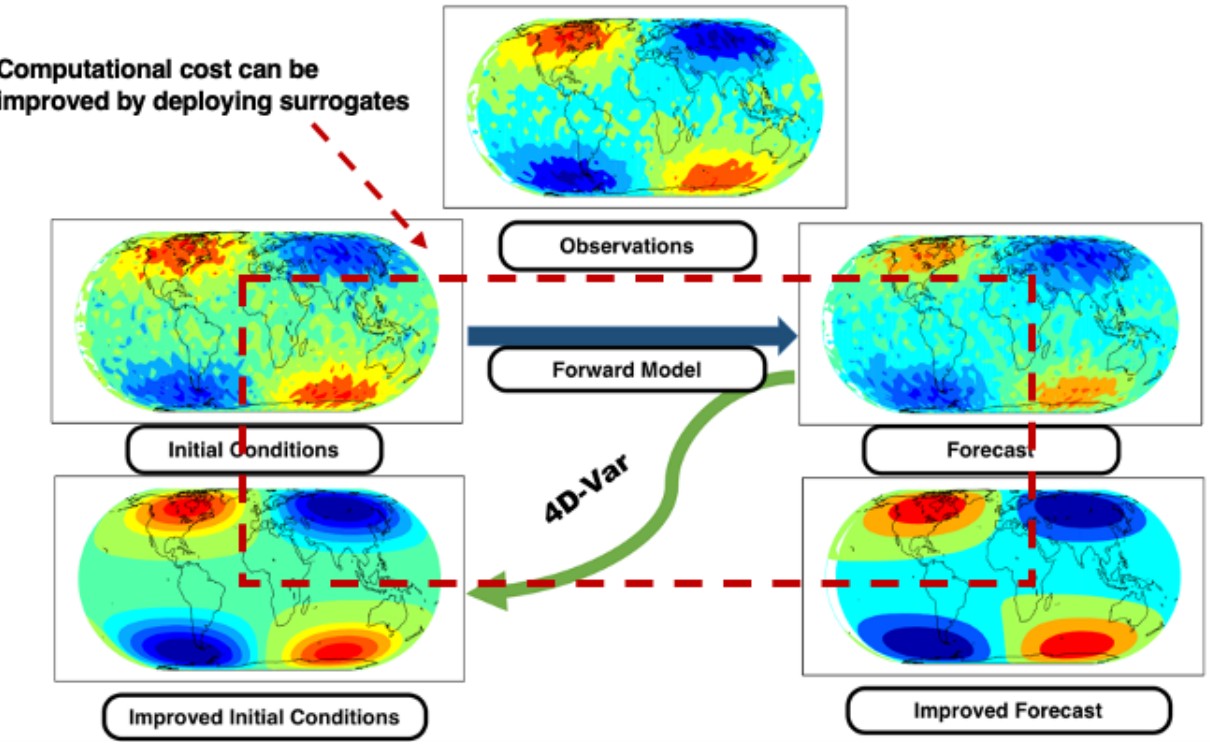

**Figure 2.** A schematic for 4D-Var. Performing 4D-Var improves the initial conditions of the system, which in turn improves the forecasts. However, 4D-Var requires simulating the expensive model multiple times and this cost can be mitigated by deploying surrogates.

comes with a background error covariance matrix $\mathbf{B}_0$. The 4D-Var method obtains an estimate $\mathbf{x}_0^{\mathrm{a}}$ of the true initial conditions by solving the following optimization problem

$$\mathbf{x}_0^{\mathrm{a}} = \arg\min_{\mathbf{x}_0} \; \mathcal{J}(\mathbf{x}_0) \qquad \text{subject to (6),} \tag{8a}$$

$$\mathcal{J}(\mathbf{x}_0) = \frac{1}{2}\|\mathbf{x}_0 - \mathbf{x}_0^{\mathrm{b}}\|_{\mathbf{B}_0^{-1}}^2 + \frac{1}{2}\sum_{i=1}^{T}\|\mathcal{H}(\mathbf{x}_i) - \mathbf{y}_i\|_{\mathbf{R}_i^{-1}}^2. \tag{8b}$$

The first term of the sum (8b) captures the mismatch of the solution $\mathbf{x}_0$ from the background $\mathbf{x}_0^{\mathrm{b}}$ at $t_0$. The second term measures the discrepancy between the forecast trajectory (model solutions $\mathbf{x}_i$) and partial observations $\mathbf{y}_i$ at future times $t_i$ in the assimilation window. The covariance matrices $\mathbf{B}_0$ and $\mathbf{R}_i$ are usually specified a-priori, and their quality influences the accuracy of the resulting solution.





## 3.2 Data assimilation with a surrogate model

Given a low-dimensional differentiable surrogate model, for instance the proposed POD-LSTM, that approximates the dynamics of the full-order numerical model, the variational DA process can be accelerated dramatically. If such a surrogate model has predicted $\hat{\boldsymbol{x}}_1$ to $\hat{\boldsymbol{x}}_T$ for a range of timesteps in the reduced-space (i.e., $\hat{\boldsymbol{x}} \approx P\mathbf{x}$, $\hat{\boldsymbol{x}} \in \mathbb{R}^K$, with $K \ll N$) and observations

from the real system, $\mathbf{y}_1$ to $\mathbf{y}_T$ are available, the objective function of the DA problem may be obtained by reconstructing the model state from the compressed representation. Here, $P : \mathbb{R}^N \to \mathbb{R}^K$ is the function that maps from physical space to reduced space. In case of POD based compression $P$ is a linear operation that projects from the physical space to the basis spanned by the truncated set of the left singular vectors of an SVD performed on snapshots from full-state model evaluation. If inversion from the truncated subspace is represented as $P^\dagger$, this gives us

$$\mathcal{J}(\hat{\boldsymbol{x}}_\mathbf{0}) = \frac{1}{2}\|\hat{\boldsymbol{x}}_\mathbf{0} - P\mathbf{x}_0^{\mathrm{b}}\|_{\mathbf{B}_0^{-1}}^2 + \frac{1}{2}\sum_{i=1}^{T}\|\mathcal{H}(\hat{\mathbf{x}}_i) - \mathbf{y}_i\|_{\mathbf{R}_i^{-1}}^2. \tag{9}$$

Note that $\hat{\mathbf{x}}_i$, which is an approximation to the state at time $t_i$, is evaluated using the surrogate model—that is, by propagating $\hat{\boldsymbol{x}}_\mathbf{0}$ using the surrogate model (LSTM in this case) and applying $P^\dagger$ to the result. This can be written mathematically as

$$\hat{\mathbf{x}}_i = P^\dagger \boldsymbol{h}(\hat{\boldsymbol{x}}_\mathbf{0}). \tag{10}$$

Given a fixed $P^\dagger$ with an effective compression ratio (obtained from our dimensionality reduction technique), Equation 8b

becomes a cost function (9) expressed in $K$ dimensions which is amenable to rapid updates of the initial conditions. Moreover, gradients of this objective function with respect to the initial conditions are trivially computable because of the use of automatic differentiation of our LSTM neural network. The overall approach is as follows:

- For each forecast window, collect random observations from the true state.

- Perform an optimization of Equation 9 by perturbing the inputs to the LSTM neural network. This input is the projected
version of the initial condition state.

- If optimization has converged, perform one forecast with the optimized initial condition. This is the DA improved forecast.

The optimization methodology used in this article is the sequential least-squares programming approach Nocedal and Wright (2006) implemented in SciPy. The neural architecture was deployed using TensorFlow 2.4. Our overall implementation is in

Python.

## 4 Results

We describe the dataset used in our experiments and then present our experimental results.





## 4.1   Dataset

The data used in this study is a subset of 20 years of output from the regional climate model WRF version 3.3.1, prepared
with methods and configurations described by  Wang and Kotamarthi (2014). WRF is a fully compressible and nonhydrostatic
regional numerical prediction system with proven suitability for a broad range of applications. The WRF simulations used
by this study are driven by reanalysis data NCEP-Reanalysis2 (NCEP-R2) for the period 1984–2003. The NCEP-R2 dataset
assimilates many of the observational datasets available to build dynamically consistent gridded fields that are typically used
for initialization and boundary condition setting for forecasting models. The WRF simulation domain is centered at 52.24°N
and 105.5°W and has dimensions of 600×516 horizontal grid points in the west-east and south-north directions with a grid
spacing of 12 km, covering most of North America. Spectral nudging technique is applied for zonal and meridional winds,
temperautre, and geopotential height at each vertical level above 850 hPa (e.g., around 1.5km).

We use geopotential height of the 500 hPa pressure surface (Z500 hereafter) from WRF output to demonstrate the approach
developed in this study. The geopotential height represents the height of pressure surface above sea level; for Z500, it is around
5.5 km, and it is often referred to as a steering level. The weather systems beneath, near to the Earth's surface, roughly move
in the same direction as the winds at the 500 hPa level. This is also the level where one can look for vertical motions. Z500
has been used as one of the standard fields for weather forecasting because it does not have very strong local gradients (in
contrast to fields such as humidity or precipitation) and does not depend on local conditions such as topography, yet most of
the important global flow patterns—such as midlatitude jets and a gradient between poles and Equator—are visible in Z500.
Low Z500 values indicate troughs and cyclones (e.g., favorable to precipitation) in the middle troposphere while high Z500
values indicate ridges and anticyclones.

We retrieve the Z500 data from the WRF output at a grid spacing of 12km and at 3 hour intervals with 515×599 grid cells in
the south-north and west-east directions, respectively. We then calulate per-grid-point daily averages to obtain one data value
per grid point per day. Spatially, we coarsen the data by five strides which still maintains the spatial patterns of Z500 but
reduces the data size significantly. Each daily Z500 snapshot therefore comprises 102×119 60km×60km grid points. We have
3287 such daily snapshots for the period 1 January 1984 to 31 December 1991. Despite the coarsening of the original 12km
data, the typical eastward propagating waves are clearly visible in the northern hemisphere. Such waves are also observed in
the real atmosphere and are one of the main features of midlatitude weather variability on timescales of several days.

We use the first six years of the daily averaged Z500 WRF data (1984–1989) for surrogate training and optimization. Of those
data, we select 70% at random as *training data*, for use in training the supervised ML formulation, and keep the remaining 30%
as *validation data*, for use in tuning the neural architecture hyperparameters and to control overfitting (a situation in which the
trained network predicts well on the training data but not on the test data). We also construct a *test data* set consisting of one
year of records (1991) for prediction and evaluation. We skip 1990 so as to ensure that there is no overlap in input windows. We
note that we choose identity as our observation error covariance matrix and we use a scaling to improve convergence. In our
future studies, in addition to using a realistic observation error covariance matrix, we will also pursue using a flow-dependent
background error covariance matrix as described in Buehner (2005).





## 4.2 Experiments

We conduct a number of experiments to evaluate the performance of our emulator-assisted DA approach. In the first, we perform a grid-based search in which we vary both the number of POD modes (5, 10, 15) and the size of the input window (7, 14, 28, or 42 days of lead-time information) to identify the optimal combination for forecasting. Other hyperparameters such as learning rate, learning rate scheduler, number of LSTM cells, and number of neurons are determined by a combination of experience in previous modeling tasks, considerations of computational efficiency, and limited manual tuning. We set the output forecast window to 20 days to represent a difficult forecast task for the emulator.

We give the complete set of hyperparameters for our problem in Table 1. We found that the lowest validation errors were achived when just five modes were retained, a result that matches earlier studies Lario et al. (2021), where increasing the number of modes is seen to cause difficulties in long-term forecasting (for example, for more than two weeks). The training and validation data are used in this phase of our experimentation to identify the best model for performing accelerated DA. We train the neural network architecture by penalizing the loss function on the training data, while using the validation data to enforce an early stopping criterion (i.e., prevent overfitting). Once the different models have been trained, the best model is determined by studying the validation losses of all the different architectures. This model is then used for testing on unseen data and for DA. For a further sensitivity analysis of this model, we trained four other models with varying input window sizes (7, 28, 35, 42) and different random seeds, with other hyperparameters fixed, and utilized them for obtaining ensemble test results. We trained and validated architectures using multiple hyperparameters in parallel by using Ray Moritz et al. (2018), a TCP/IP-based parallelization protocol, on Nvidia A100 GPUs of the Argonne Leadership Computing Facility's Theta supercomputer.

**Table 1.** LSTM hyperparameters used when training the low-dimensional surrogate. The asterisked quantity is varied, in addition to random seeds, for sensitivity analyses of the surrogate model where in addition to the best chosen model here, several other models are trained with different weight initializations and input window sizes.

| | |
|---|---|
| Neurons per cell | 20 |
| Number of stacked cells | 2 |
| Initial learning rate | 1.0e-3 |
| Learning rate decay rate | 0.5 (based on 10 epochs patience) |
| Activation function | ReLU |
| Input window* | 7, 14, 28, or 42 days |
| Number of modes retained | 5 |
| Output window | 20 days |
| Weight initialization | Glorot |
| Optimizer | Adam |

Our first set of assessments test the POD-LSTM emulator without any DA. We show in Figure 3 how emulator predictions for day 15 (for all forecasts in the test region) compare to climatology. Climatology, here, refers to the average forecast on





a specific forecast day based on averaged observations from the training data. For instance, if geopotential height is to be forecast on December 7, 1991, the climatology prediction would be the average of December 7 geopotential height values for 1984-1989. The metrics we use for comparison first include the Cosine similarity improvement as shown in Figure 3(a). Here,

the cosine similarity (which captures the alignment between prediction and truth) obtained from climatology is subtracted from that obtained from the emulator forecast. Thus, in this case, negative (blue) regions are where climatology captures the temporal trend of the forecast better than our emulator. In Figure 3(b), we subtract the MAE of the emulator predictions from that of climatology. Here, the blue regions are where climatology is more accurate on average than the emulator. Figure 3(c) merely shows the MAE for the emulator. From this analysis, it is clear that for large regions in the data domain (particularly in

the North), the emulator performs quite poorly.

We next assess results from applying DA through random observations at 5000 locations of the domain, to see if results obtained with the emulator can be improved. Here, sparse observations at each timestep of the output window are used within the optimization statement introduced in Equation 9 to update the initial conditions (i.e., the input window). We emphasize that the observations are obtained randomly from the true state of the system during forecasts, which corresponds to the test data

introduced previously.

The results, in Figure 4, show that the use of sensor information aids the forecast immensely, improving performance in all metrics significantly by virtue of DA. In particular, the proposed augmentation (i.e., the use of DA during forecasts) reduces forecast errors considerably in regions where the sole use of the emulator was not competitive. We provide MAE assessments for our 20 day output averaged over the testing data range in Figure 5, where we compare the raw emulator outputs, climatology,

persistence, and the DA-corrected outputs. We also provide confidence intervals for the regular (i.e., without DA) and DA-corrected emulators where the five different emulators are trained with different random seeds and input window durations to assess the sensitivity to the initialization of the training as well as the automatic differentiation based optimization. The results indicate that the DA-corrected emulation is the most accurate and consistent forecast technique.

We also provide in Figure 6 a comparison with another benchmark, persistence, which uses the state of the last day in the

input window as the forecast for each day of the output window. Persistence is seen to be more accurate for short lead times but is outperformed by both climatology and data-driven methods for longer forecast durations.

Our choice of 20 days for the forecast window was motivated by the results of Rasp et al. Rasp et al. (2020); Rasp and Thuerey (2021), who performed analyses for five-day forecasts. Our study increased the forecast window to four times the original assessments for geopotential height to demonstrate the possibility for improved forecasting of *any emulator* using

automatic differentiation-enabled variational DA. In many subregions of the computational domain, the use of DA with the emulator is also seen to improve on climatology predictions—even at large lead times (20 days out), where classical data-driven and numerical forecast models are typically less accurate than climatology.

Our method also delivers large computational gains. A one year forecast with the emulator-assisted DA performed each day takes approximately one hour on a single processor core without any accelerator hardware. In contrast, the original PDE-based

simulation required 21,600 core hours for a one-year forecast with a 515×599 grid. (A coarse-grained forecast for one year on the 103×120 grid used here takes 172 core hours, but the resulting flow field does not adequately reproduce the fidelity of the





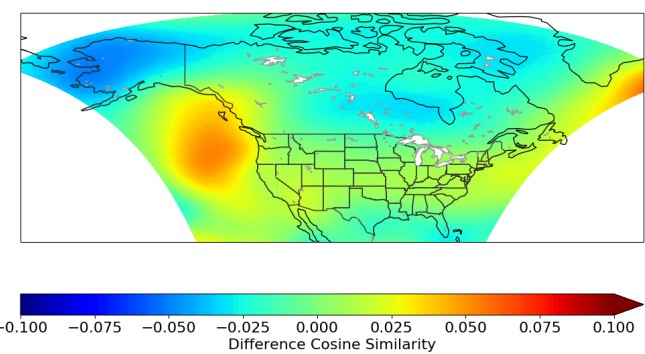

(a) Similarity improvement: Regular emulator without DA

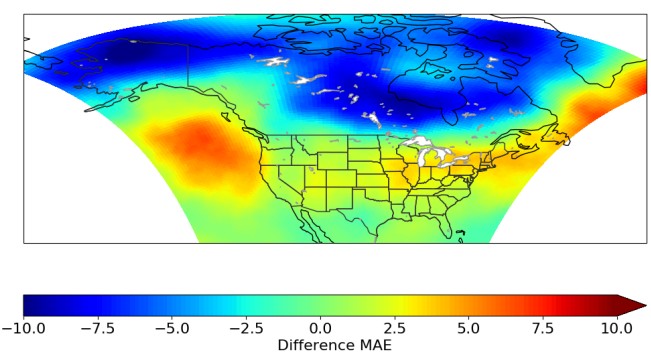

(b) MAE improvement: Regular emulator without DA

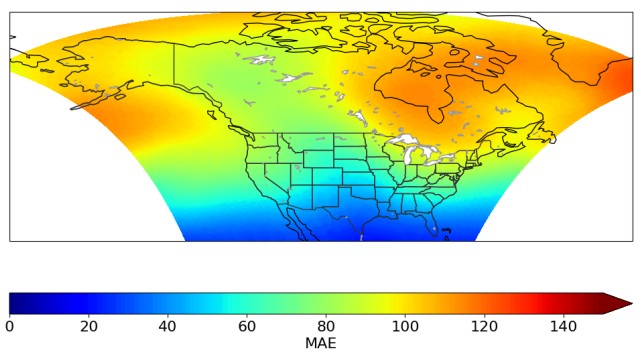

(c) MAE: Regular emulator without DA

**Figure 3.** Improvements over climatology for regular emulator, when using the best set of hyperparameters as determined on validation data. Here cosine similarity captures an agreement in the trend between forecast and truth and MAE refers to the mean absolute error between the same.



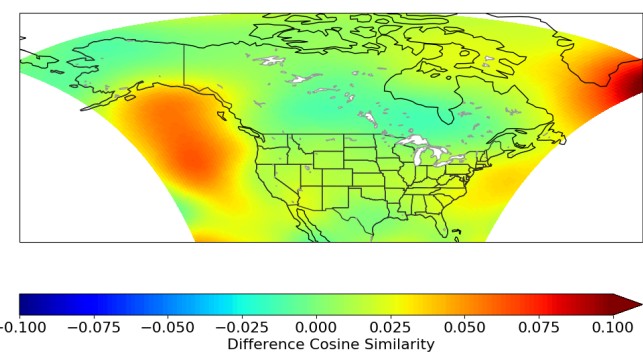

(a) Similarity improvement: Regular emulator + DA

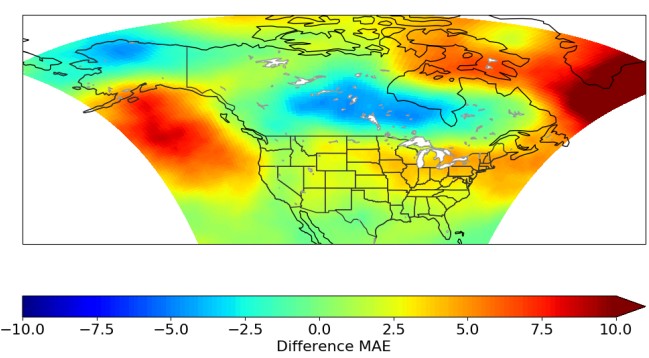

(b) MAE improvement: Regular emulator + DA

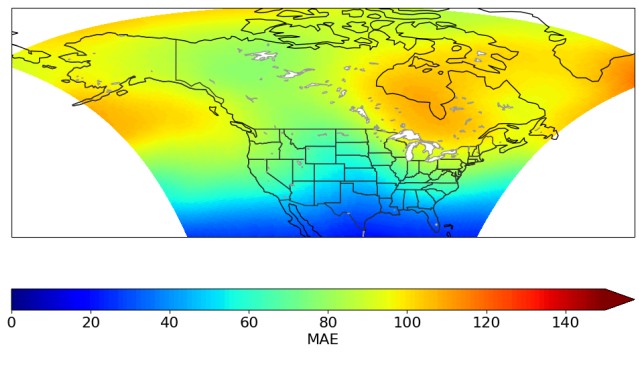

(c) MAE: Regular emulator + DA

**Figure 4.** Improvements over climatology for regular emulator corrected with DA, when using the best set of hyperparameters as determined on validation data. Here cosine similarity captures an agreement in the trend between forecast and truth and MAE refers to the mean absolute error between the same.




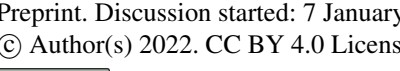

(a)  Alaska

(b)  Midwest

(c)  North Great Plains

(d)  Northeast

(e)  Northwest

(f)  South Great Plains

(g)  Southeast

(h)  Southwest

(i)  North America

**Figure 5.** 20 day geopotential height forecast MAEs, relative to the true test data, for seven National Climate Assessment subregions of continental North America Reidmiller et al. (2018), from both a regular ML emulator and the same emulator corrected by variational DA. Confidence intervals, for five separately trained emulators, encapsulate the effects of perturbations to the random seed and the input window. DA-based correction gives both lower MAEs and narrower confidence intervals.



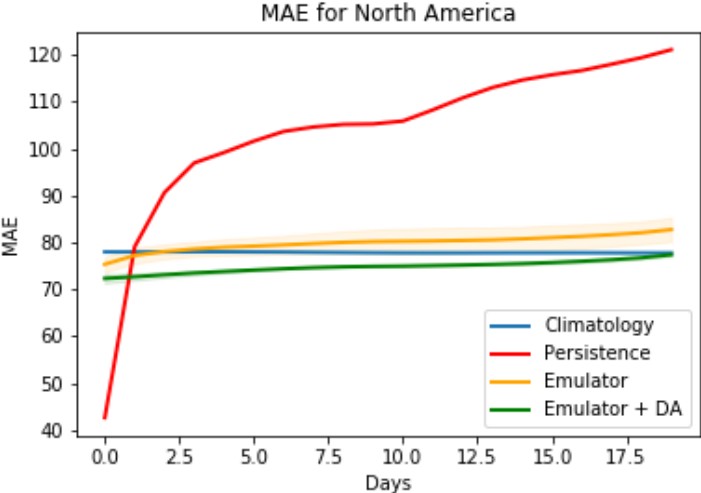

**Figure 6.** 20 day geopotential height forecast MAEs for continental North America, relative to the true test data. Results describe performance of a regular ML emulator and the same emulator corrected by variational data assimulation. Here we also show the results of persistence which is outperformed in the long term prediction regime as expected.

515×599 grid.) One can observe significant speed up ($\sim 10^4$ times) for the emulation of the geopotential height, even without factoring the cost of an additional variational DA step. Furthermore, the solution to the 4D-Var problem, which yields an improved initial condition, requires on the order of 100 model runs, where each model run can be several orders of magnitude
more expensive than our emulator.

## 5  Conclusions

We have described how a differentiable reduced-order surrogate geophysical forecasting model may be integrated into an outer-loop optimization technique whereby real-time observations of the true solution are used to improve the forecast of the surrogate. We use such observations to improve the initial condition of the surrogate model, such that an optimization statement
given by the classical 4D variational DA objective function is minimized. The use of the reduced-order surrogate converts a high-dimensional optimization to one that is given by the dimensionality of the compressed representation and the duration of the input window for forecasting. Our optimization is thus performed rapidly, given sparse and random observations from the true flow field, without any access to high performance computing resources. Computational costs are reduced by four orders of magnitude, by virtue of the surrogate assisted forecasting and variational DA, when compared to a classical equation-based
forecasting of the dynamics. We assess our model on a real-world forecasting task for geopotential height in the continental US, and obtain competitive results with respect to climatology and persistence baselines for mean-absolute-error and cosine similarity.



*Code and data availability.* The data that support the findings of this study are available from the corresponding author upon reasonable request.

*Author contributions.* RM designed study, wrote code, performed experiments, analyzed results; VR and EC developed the variational data assimilation; JW and RK developed the model output used for developing the surrogate; PB and BL assisted with the development of surrogate model and algorithms and IF participated in the model analysis and development of the manuscript.

*Competing interests.* There are no competing interests

*Acknowledgements.* This material is based upon work supported by Laboratory Directed Research and Development (LDRD) funding from
345 Argonne National Laboratory, provided by the Director, Office of Science, of the U.S. Department of Energy under Contract No. DE-AC02-06CH11357. This material is partially based upon work supported by the U.S. Department of Energy (DOE), Office of Science, Office of Advanced Scientific Computing Research, under Contract DE-AC02-06CH11357. This research was funded in part by, and used resources of, the Argonne Leadership Computing Facility, a DOE Office of Science User Facility supported under Contract DE-AC02-06CH11357. Romit Maulik acknowledges the support of ASCR Project DOEFOA2493: 'Data intensive scientific machine learning'. Gianmarco Mengaldo
acknowledges support from NUS startup grant R-265-000-A36-133.



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
