# Peer review of "AIEADA 1.0: Efficient high-dimensional variational data assimilation with machine-learned reduced-order models"

_Geoscientific Model Development, 2021_

## Referee Comment (RC1)

**Review of gmd-2021-415: *AIEADA 1.0: Efficient high-dimensional variational data assimilation with machine-learned reduced-order models**

By Romit Maulik, Vishwas Rao, Jiali Wang, Gianmarco Mengaldo, Emil Constantinescu, Bethany Lusch, Prasanna Balaprakash, Ian Foster and Rao Kotamarthi

18 Feb 2022

Verdict: **Major Revision**

In this manuscript the authors create a machine-learning based emulator of a dynamical model and then integrate it into a standard 4D-Var data assimilation algorithm. They employ a long short-term memory (LSTM) neural network as the emulator, and demonstrate its advantages for 4D-Var, thanks to its easy differentiability. The emulator is demonstrated on a data set produced from WRF simulations, which is impressive given that many of these first studies only consider toy models. The individual components of this system have been tested before by others, but as far as I am aware, the combination shown here is novel.

The paper is mostly well written, and the authors are to be commended especially for the introduction which presents an excellent overview not only of data assimilation algorithms but also the latest results from machine-learning/data assimilation hybrid studies. I would be very happy to see this paper published. However, I did find certain aspects of the description of the techniques and experiments to be lacking. In particular, I did not find the explanation of the LSTM system to be sufficient for me, as a non-expert, and it left me with many unanswered questions. I would advise the authors to go back and rethink this part from the perspective of a data assimilation person. My detailed advice is given below, but in summary: try to explain in plain language what it is that an LSTM neural network is aiming to do, how it compares with the traditional numerical models that it is intended to replace here (from an "external interface" perspective — can an LSTM be a drop-in replacement for a numerical model?), and how it is integrated into a traditional data assimilation system. I have also provided some minor comments which may improve the readability of the paper.

I look forward to the authors' revision.

**Major Comments**

- Section 2.2: As someone who does not use LSTMs, I did not understand any of this section I'm afraid (except for the final five lines). After reading many times, I am still not sure what problem the LSTM is designed to solve. If the authors would like this paper to be read and understood by a broader audience, I think it would be wise for the authors to revise this section substantially. The following tips may be considered. They may seem pedantic, but when we are precise we eliminate confusion.

- ○ The LSTM is introduced as a forecasting technique, in which case I would expect to see an equation like x_t+1 (future state) = M(x_t) (operator applied to initial condition). Yet I cannot discern any such expression from those given. How does one take the Equations 4 and 5 and actually get a future state from an initial state?
  - ○ There are many undefined terms and variables, including r, t, T, input sequence, input window, functional prediction, cell and output window.
  - ○ The term "observation" on line 157 is ambiguous. I would suggest another term in case the reader thinks of actual meteorological observations.
  - ○ What is $\mathbf{h}$? Is it a vector? If so, what dimensions does it have? What does the t subscript mean, and why are all $\mathbf{h}$'s in Equation 5 the same?
  - ○ What about $h$? On line 160 it says that this is the output of the function approximated by the LSTM — does that mean $h$ is the LSTM itself?

**Minor Comments**

- Lines 24, 66: Slight typo — *uncertainties.*
- Line 113: Two "with"s.
- Line 139: I know it's the title of the section, but could the authors define the POD acronym here anyway?
- Line 146: For the benefit of readers (including myself) who have not encountered SVD for a while, could the authors define all of the terms, U, Σ and V here, including their dimensions? E.g., please mention that the columns of U and V are the left- and right-singular vectors, respectively, and that the diagonal of Σ contains the singular values. Additionally, this may be my misunderstanding, but please could the authors double check the stated dimensionality of U? My understanding was that the U and V matrices are square.
- Line 188: Typo — *references.*
- Line 195: Please specify *Figure* 2, just in case the reader looks to Section 2.
- Lines 216-217: This caused me confusion for a long time because I didn't notice the lack of italics in $\hat{\text{x}}_i$. I thought that $\hat{\text{x}}_i$ and $\hat{x}_0$ were in the same vector space, when in fact $\hat{\text{x}}_i$ is in $\Re^N$ and $\hat{x}_0$ is in $\Re^K$. Could the authors clarify the notation to make Equations 9 and 10 easier to comprehend? For example, perhaps all hat'ed variables can be in the reduced-space?
- Line 236: Did the authors mean to say regional *numerical* weather prediction system?
- Line 242: Typo — *temperature.*
- Line 253: Typo — *calculate.*
- Line 254: What does it mean to "coarsen the data by five strides"? Are the coarsened fields obtained by averaging, or simply by subsetting the high-resolution fields?
- Line 264: Z500 is around 5,000 m, so if the identity is chosen as the observation error covariance matrix, that must mean that the simulated errors are on the order of 1 m, which is negligible. Is it definitely the identity, or did the authors mean that it is the identity multiplied by a constant observation error (perhaps 50 m or something)?

- Line 264: Also, please explain for the uninitiated the significance of using the identity matrix as the observation error covariance matrix: i.e., this means that the observation errors are uncorrelated.
- Line 275: Typo — *achieved.*
- Line 285: Are the results presented here (and in Figure 3) averaged across all of the forecasts performed? Or are we looking at one specific case, as representative of all of the cases?
- Line 286: "For all forecasts in the test region" — the test data set covers the year 1991, but how are the forecast periods constructed? If the "output window" is 20 days (meaning the forecast is performed out to 20 days), are the forecasts performed back-to-back, meaning there are only 365/20 ≈ 18 forecasts in total? Or do the forecast intervals overlap, so there are 365 forecasts — one for each day?
- Line 286: Should this be "test data set" rather than "test region"? "Region" implies some kind of spatial meaning.

---

## Author Comment (AC2)

**Response letter**

**Reviewer 1**

*In this manuscript the authors create a machine-learning based emulator of a dynamical model and then integrate it into a standard 4D-Var data assimilation algorithm. They employ a long short-term memory (LSTM) neural network as the emulator, and demonstrate its advantages for 4D-Var, thanks to its easy differentiability. The emulator is demonstrated on a data set produced from WRF simulations, which is impressive given that many of these first studies only consider toy models. The individual components of this system have been tested before by others, but as far as I am aware, the combination shown here is novel.*

*The paper is mostly well written, and the authors are to be commended especially for the introduction which presents an excellent overview not only of data assimilation algorithms but also the latest results from machine-learning/data assimilation hybrid studies. I would be very happy to see this paper published. However, I did find certain aspects of the description of the techniques and experiments to be lacking. In particular, I did not find the explanation of the LSTM system to be sufficient for me, as a non-expert, and it left me with many unanswered questions. I would advise the authors to go back and rethink this part from the perspective of a data assimilation person. My detailed advice is given below, but in summary: try to explain in plain language what it is that an LSTM neural network is aiming to do, how it compares with the traditional numerical models that it is intended to replace here (from an "external interface" perspective — can an LSTM be a drop-in replacement for a numerical model?), and how it is integrated into a traditional data assimilation system. I have also provided some minor comments which may improve the readability of the paper.*

*I look forward to the authors' revision.*

**Response:** We thank the Referee for their support of our research. In the following, we have outlined our responses and relevant changes in the manuscript to address their concerns.

**Major comments:**

*Section 2.2: As someone who does not use LSTMs, I did not understand any of this section I'm afraid (except for the final five lines). After reading many times, I am still not sure what problem the LSTM is designed to solve. If the authors would like this paper to be read and understood by a broader audience, I think it would be wise for*

*the authors to revise this section substantially. The following tips may be considered. They may seem pedantic, but when we are precise we eliminate confusion.*

**Response:** We have incorporated the suggestions as outlined below.

*The LSTM is introduced as a forecasting technique, in which case I would expect to see an equation like x_t+1 (future state) = M(x_t) (operator applied to initial condition). Yet I cannot discern any such expression from those given. How does one take the Equations 4 and 5 and actually get a future state from an initial state?*

**Response:** We have added extensive explanation regarding the LSTM neural architecture that improves the presentation of Section 2.2. Specifically, we have introduced all the linear algebra operations that occur within the LSTM architecture which allow it to forecast the state of a dynamical system with both short and long term correlations. We have also clarified what 'encoding' refers to in the context of our architecture, which is the process by which information is extracted and compressed from a window of inputs and utilised for subsequent forecasting. All variables are now defined for the convenience of the reader.

*There are many undefined terms and variables, including r, t, T, input sequence, input window, functional prediction, cell and output window.*

**Response:** We have updated the presentation of the LSTM neural architecture to address these terms.

*The term "observation" on line 157 is ambiguous. I would suggest another term in case the reader thinks of actual meteorological observations.*

**Response:** We have used "historical data" instead of observations.

*What is h? Is it a vector? If so, what dimensions does it have? What does the t subscript mean, and why are all h's in Equation 5 the same?*

**Response:** h is a vector and its dimensions are given by the number of hidden layer neurons, t refers to time and all the values of h in Equation 5 are the same to indicate that the encoded value of the input window is the input at each time step of the output window. Additional clarifications were included in Sec. 2.2.

*What about h? On line 160 it says that this is the output of the function approximated by the LSTM — does that mean h is the LSTM itself?*

**Response:** We hope that the revision has made this clear. The symbol h refers to the encoded representation of the input window.

**Minor Comments**

Lines 24, 66: Slight typo — uncertainties.

**Response:** This has been fixed.

Line 113: Two "with"s.

**Response:** This has been fixed.

Line 139: I know it's the title of the section, but could the authors define the POD acronym here anyway?

**Response:** We have identified the POD acronym in the title of Section 2.1, and have redefined POD in the text here:
"Here, we use a projection …. subset of the *proper orthogonal decomposition (POD) basis*".

Line 146: For the benefit of readers (including myself) who have not encountered SVD for a while, could the authors define all of the terms, U, Σ and V here, including their dimensions? E.g., please mention that the columns of U and V are the left- and right-singular vectors, respectively, and that the diagonal of Σ contains the singular values. Additionally, this may be my misunderstanding, but please could the authors double check the stated dimensionality of U? My understanding was that the U and V matrices are square.

**Response:** We have expanded on the explanation of the SVD - which in this case refer to the compact variant. Here U, V are semi-unitary matrices which need not be square. We have mentioned this as follows:

"Note that due to the compact nature of the SVD, U and V are semi-unitary matrices which need not be square"

Line 188: Typo — references.

**Response:** This has been fixed.

Line 195: Please specify Figure 2, just in case the reader looks to Section 2.

**Response:** This has been fixed.

Lines 216-217: This caused me confusion for a long time because I didn't notice the lack of italics in $\hat{\text{x}}_i$. I thought that $\hat{\text{x}}_i$ and $\hat{x}_0$ were in the same vector space, when in fact $\hat{\text{x}}_i$ is in and $\mathfrak{R}N$

$\hat{x}_0$ is in . Could the authors clarify the notation to make Equations 9 and $\Re K$ 10 easier to comprehend? For example, perhaps all hat'ed variables can be in the reduced-space?

**Response:** Thank you for this suggestion. We have updated the notation in Section 3.2 to ensure consistency.

Line 236: Did the authors mean to say regional numerical weather prediction system?

**Response:** That is correct - we have updated this in the revision.

Line 242: Typo — temperature.

**Response:** This has been fixed.

Line 253: Typo — calculate.

**Response:** This has been fixed.

Line 254: What does it mean to "coarsen the data by five strides"? Are the coarsened fields obtained by averaging, or simply by subsetting the high-resolution fields?

**Response:** Here, we mean to imply subsampling, evenly, the high-resolution field to generate our target resolution. We added a brief clarifying point in the manuscript.

Line 264: Z500 is around 5,000 m, so if the identity is chosen as the observation error covariance matrix, that must mean that the simulated errors are on the order of 1 m, which is negligible. Is it definitely the identity, or did the authors mean that it is the identity multiplied by a constant observation error (perhaps 50 m or something)?

**Response:** We have added text to indicate the settings used for our numerical experiments. The standard deviation of the noise we use is approximately 1.5% of the mean.

Line 264: Also, please explain for the uninitiated the significance of using the identity matrix as the observation error covariance matrix: i.e., this means that the observation errors are uncorrelated.

**Response:** We have added text indicating that we assume uncorrelated noise. We also note that the noise can be correlated in certain instruments and using a correlated observation error matrix is a subject of future research.

Line 275: Typo — achieved.

**Response:** This has been fixed.

Line 285: Are the results presented here (and in Figure 3) averaged across all of the forecasts performed? Or are we looking at one specific case, as representative of all of the cases?

**Response:** These results are averaged across all the forecasts performed. We have added this to the captions of the relevant figures.

Line 286: "For all forecasts in the test region" — the test data set covers the year 1991, but how are the forecast periods constructed? If the "output window" is 20 days (meaning the forecast is performed out to 20 days), are the forecasts performed back-to-back, meaning there are only 365/20 ≈ 18 forecasts in total? Or do the forecast intervals overlap, so there are 365 forecasts — one for each day?

**Response:** The forecast intervals overlap - so one forward forecast for 20 days of output for each day (so a total of 365 forecasts as the Referee suggests).

Line 286: Should this be "test data set" rather than "test region"? "Region" implies some kind of spatial meaning.

**Response:** The reviewer is correct - this should be "test data set". We have fixed it.

---

## Author Comment (AC3)

**Reviewer 2**

*Excellent paper. Great improvement from previous studies where the ML approach learns only the misfits whilst the authors have implemented an on-the-fly system.*

**Response:** We thank the Referee for their support of our work.

*Figure 3 and 4 need to be together for a better comparison.*

**Response:** We have made this change.

*Architectures of networks used?*

**Response:** We have added extensive explanation related to the LSTM architecture used in the revision.

*Was layer normalisation used? Why not?*

**Response:** Layer normalisation was not explored in this work. We note that the main focus of our study was to show that an inaccurate surrogate can be improved by observations during deployment in contrast with explicitly finding the best possible surrogate. We believe several methodological tricks can be used to potentially improve our LSTM implementation.

*What other temporal NN could be used? Transformers?*

**Response:** Methods that are in the family of recurrent neural networks would be appropriate for this forecasting. Some examples are transformers, neural ordinary differential equations, gated recurrent units. We note that while several different surrogate modeling strategies are viable for the forward model, the purpose of this article was to demonstrate data assimilation with a fixed model with improved results.

*What is the effort of using a CNN instead of a PCA reduction to feed the LSTM AE?*

**Response:** We decided to utilise PCA instead of CNN for maximising the interpretability of the latent space although this would lead to reduced compression efficiency. Several recent articles have looked at autoencoders for dimensionality reduction and our data-assimilation strategy is equally applicable for that scenario. We have added a statement to clarify this in the main article.

*I would like to see a brief discussion of how an adversarial approach might help the rollout of the forecast here.*

**Response:** We thank the reviewer for this suggestion. In this work, we have not explored adversarial training of the LSTM, but suspect it would be helpful in making the forecasts more robust to noisy inputs. A prominent failure mode of RNN architectures is the growth of errors due to the autoregressive nature of forecasting (small errors compounding to grow uncontrollably) and this approach may assist with that. We shall be exploring this for our future studies.

---

## Author Response (AR2)

We have added affiliation information and information about our code DOI (through zenodo) in
the document. Thank you.